# The Gene *vepN* Regulated by Global Regulatory Factor *veA* That Affects Aflatoxin Production, Morphological Development and Pathogenicity in *Aspergillus flavus*

**DOI:** 10.3390/toxins16040174

**Published:** 2024-04-03

**Authors:** Jia Xu, Mengqi Jiang, Peng Wang, Qing Kong

**Affiliations:** 1School of Food Science and Engineering, Ocean University of China, Qingdao 266404, China; xuj970719@163.com (J.X.); jmengqi3747@163.com (M.J.); 2College of Marine Life Science, Ocean University of China, Qingdao 266003, China; wangp960517@163.com

**Keywords:** aflatoxin, VeA, Aspergillus flavus, *vepN*, conidiation, sclerotia, pathogenicity

## Abstract

Velvet (VeA), a light-regulated protein that shuttles between the cytoplasm and the nucleus, serves as a key global regulator of secondary metabolism in various *Aspergillus* species and plays a pivotal role in controlling multiple developmental processes. The gene *vepN* was chosen for further investigation through CHIP-seq analysis due to significant alterations in its interaction with VeA under varying conditions. This gene (AFLA_006970) contains a Septin-type guanine nucleotide-binding (G) domain, which has not been previously reported in *Aspergillus flavus* (*A. flavus*). The functional role of *vepN* in *A. flavus* was elucidated through the creation of a gene knockout mutant and a gene overexpression strain using a well-established dual-crossover recombinational technique. A comparison between the wild type (WT) and the Δ*vepN* mutant revealed distinct differences in morphology, reproductive capacity, colonization efficiency, and aflatoxin production. The mutant displayed reduced growth rate; dispersion of conidial heads; impaired cell wall integrity; and decreased sclerotia formation, colonization capacity, and aflatoxin levels. Notably, Δ*vepN* exhibited complete growth inhibition under specific stress conditions, highlighting the essential role of *vepN* in *A. flavus*. This study provides evidence that *vepN* positively influences aflatoxin production, morphological development, and pathogenicity in *A. flavus*.

## 1. Introduction

*Aspergillus flavus* (*A. flavus*), a common saprophytic fungus and an opportunistic pathogen, is prevalent in soil and air, reproducing by both asexual and sexual means [1,2]. It not only infects various economically significant crops, such as peanuts, corn, cotton, and rice, but also ranks among the most prevalent *Aspergillus* species causing invasive and allergic diseases in humans [3,4]. *A. flavus* contamination primarily occurs through the production of secondary metabolites, notably, aflatoxin (AF), with aflatoxin B_1_ (AFB_1_) being particularly notorious [5,6]. The biosynthesis of AF involves a complex pathway [7,8,9] regulated by intra-cluster and extra-cluster genes. Among these, *veA*, an extra-cluster gene, plays a crucial role in aflatoxin biosynthesis and was initially identified as essential for light-dependent conidiation in *Aspergillus nidulans* (*A. nidulans*) [10]. The light-associated Velvet complex, comprising the proteins VeA, LaeA, and VelB, forms a conserved regulatory unit in dimorphic and filamentous fungi, influencing the biosynthesis of diverse secondary metabolites [11,12]. VeA, predominantly expressed in the dark, physically associates with VelB, which is expressed during sexual development, bridging VelB to the nuclear master regulator of secondary metabolism, LaeA. Deletion of either *velB* or *veA* results in defects in sexual fruiting-body formation and secondary metabolite production [13]. Further investigations revealed that the *veA* mutant is incapable of producing sclerotia or mycotoxins ST, versicolorin A, AFB (aflatoxin B_1_ and aflatoxin B_2_), and aflatrem; however, conidial production remains unaffected [14,15,16,17]. In this study, a monoclonal antibody targeting VeA was developed to facilitate chromatin immunoprecipitation and sequencing (ChIP-seq). Genes closely interacting with VeA under varying conditions were identified [18], with AFLA_006970, referred to as *vepN* in this study, standing out for further exploration.

The gene *vepN* contains a Septin-type guanine nucleotide-binding (G) domain, representing a conserved protein family from yeast to humans belonging to the P-loop GTPase superfamily. Septin proteins regulate various biological processes, playing crucial roles in cell cytoskeleton, cell morphogenesis, intracellular transport of metabolites, membrane trafficking, and cell polarity, by modulating the transition between the GDP (inactive) and GTP (active) states. Septin proteins in *Candida albicans* influence cytoplasmic division and virulence [19], while in *A. nidulans*, Septin proteins impact conidial formation and participate in low-temperature and osmotic stress responses [20]. However, the functional role of *vepN* in *A. flavus* has not been explored. To investigate the effects of *vepN* on the growth, development, and aflatoxin synthesis of *A. flavus*, *vepN* gene knockout and overexpression mutants were established and elucidated at both indicator and transcriptional levels.

## 2. Results

### 2.1. Analysis of CHIP-Seq Experimental Results

The molecular weight of recombinant AfVeA was around 63 kDa, and Western blot analysis showed that the multiclonal AfVeA antibody (0.4–50 ng) specifically bound to AfVeA (Figure 1). Next, VeA antibody was used for the CHIP-seq experiment, and the experimental results were analyzed by GO analysis. Gene functional annotation is widely facilitated by the standard facts in the description of gene function and enrichment analysis [21]. The Gene Ontology (GO) analysis classifies enriched genes into three categories, as shown in Appendix A, including biological processes (BPs), molecular functions (MFs), and cellular components (CCs). A total of 66 differentially expressed genes (DEGs) were detected, with 22 in the BP category, 20 in the MF category, and 24 in the CC category. Regarding BPs (Appendix A), significantly differentially expressed genes (DEGs) were predominantly enriched in signal transduction processes, involving cellular cytoskeleton organization, Rho protein signal transduction and its regulation, microtubule nucleation, and more. For MFs (Appendix A), DEGs were mainly enriched in guanly nucleotide exchange factor activity and binding activity and manifested in glutaryl CoA dehydrogenase activity, Rho guanyl nucleotide exchange factor activity, tubulin binding, and protein binding. As for CCs (Appendix A), DEGs were notably enriched in supramolecular complexes, predominantly found in microtubule, supramolecular complex, supramolecular polymer, supramolecular fiber, and others. The gene *vepN* was enriched in MFs, exhibiting binding activity, specifically nucleotide binding, small molecule binding, and nucleoside phosphate binding.

### 2.2. Identification A. flavus NRRL3357 Gene vepN

Gene sequences were sourced from the National Center for Biotechnology Information (NCBI). The *vepN* open reading frame (ORF) spans 1269 base pairs (bp), encoding a protein of 401 amino acids. Homologous proteins were also retrieved from the NCBI, and their evolutionary relationship was analyzed by using MEGA11.0. The analysis revealed that VepN in *Aspergillus* spp. group into a cluster, with *A. flavus* and *A. oryzae* exhibiting the highest identity (99% identity and 100% query coverage) (see Figure 2A). In *A. flavus*, the P-loop NTPase domain is predominantly situated at the C terminus. Furthermore, a comparative examination of genes harboring this domain in other *Aspergillus* species revealed that in *Aspergillus niger* and *A. nidulans*, the domain is positioned at the C terminus, whereas in *Aspergillus fumigatus*, it is split into two fragments (see Figure 2B).

### 2.3. Construction of the Deletion Mutant (ΔvepN) and Overexpression (OE::vepN) Mutant

The *vepN* deletion mutant (∆*vepN*) was created using homologous recombination and confirmed via PCR analysis, as presented in Figure 3A,B, where the *vepN* ORF and the recombinant fragment were amplified from genomic DNA. Verification involved the use of specific primers, vepN-F and vepN-R, to detect the replacement of the gene *vepN* with the screening marker gene *pyrG*. In successfully recombined strains, *vepN* was not detectable, while the wild type (WT) strain showed amplification of *vepN* within the 1000 to 2000 bp range. Additional verification was performed using the primers vepN-5Fp, vepN-5Rp, vepN-3Fp, and vepN-3Rp to confirm the presence of upstream and downstream homologous sequences and the screening marker gene *pyrG*. The results of the OE::*vepN* transformant validation are presented in Figure 3C,D, showing the successful amplification of the gene *vepN* in the overexpression strain (OE::*vepN*) and its absence in the deletion mutant (∆*vepN*). The expression analysis of *vepN* by qRT-PCR indicated greater accumulation of *vepN* transcripts in the *vepN* overexpression strain compared with the wild type (see Figure 3E).

### 2.4. Effects of vepN on Morphology

Significant disparities in colony morphology were observed among the wild type (WT), Δ*vepN*, and OE::*vepN* strains cultivated on GMM, PDA, and YES media. The WT exhibited robust growth across all three media, notably thriving on YES medium, characterized by abundant yellow–green conidia, colonies that fully covered the medium, and densely packed hyphae (Figure 3A). In contrast, the colony diameter of Δ*vepN* on PDA and GMM media showed a marked reduction, accompanied by diminished pigment accumulation and sparse hyphal growth (Figure 3A,B). Interestingly, the colony morphology of the *vepN*-com strain reverted to its original state (Figure 3A,B).

### 2.5. Effects of vepN on Conidial Formation

Compared to the WT, the conidial heads of Δ*vepN* exhibited sparse, radial, branched structures with weak stalks, a morphology that was restored in the OE::*vepN* strain (Figure 4C). The quantification of conidia harvested from GMM, PDA, and YES media on days 3, 5, and 7 revealed progressive differences in conidial counts between Δ*vepN* and WT, with a 2.5-fold divergence by day 7 on GMM but insignificant variations on PDA (Figure 4D–F). Notably, on YES medium, ΔvepN consistently produced significantly fewer conidia compared with WT at all time points, with reductions of up to 60%. Regardless of the medium used, OE::*vepN* consistently showed the highest conidial counts. The RT-qPCR analysis indicated significant downregulation of both *brlA* and *abaA* at 48 h and 72 h (Figure 3G), impacting the normal growth and development of conidia.

### 2.6. Effects of vepN on Development of Sclerotia

After 7 days of cultivation, the WT generated a substantial quantity of brown, immature sclerotia, up to 518 per plate, whereas Δ*vepN* yielded only 56 per plate, and OE::*vepN* produced 784 per plate (Figure 5A,C). Extending the culture period to 14 days led to more visibly mature sclerotia, with WT displaying an abundance of black, plump granules tightly packed, contrasting with the smaller, sparsely growing sclerotia of Δ*vepN*. The WT recorded 659 sclerotia per plate, maintaining a noteworthy difference from Δ*vepN* (Figure 5B,C). Surprisingly, the sclerotia count in OE::*vepN* rebounded to 1003 per plate, surpassing both the WT and Δ*vepN*. The gene expression analyses revealed a decreasing trend in the relative levels of *nsdC*, *nsdD*, and *sclR*, which are genes associated with sclerotia production, across various time points (Figure 5D).

### 2.7. Effects of vepN on AFB

The synthesis of aflatoxin is influenced by both external and endogenous factors. Significant variations were observed in *A. flavus*’s capacity to produce aflatoxin on different substrates, with AFB only being present in the WT strain on GMM and PDA media, while this was undetected in the Δ*vepN* mutant (shown in Appendix A). The AFB content was notably lower in the Δ*vepN* mutant compared with the WT on YES medium. The quantitative analysis through HPLC revealed that the AFB_1_ content in Δ*vepN* was 2.94 μg/mL, representing only 7% of that in the WT, while the AFB_2_ content was 10% of that of the WT (Figure 6B). In the case of *vepN*-com, AFB production was restored, with AFB_1_ and AFB_2_ levels reaching 55.98 μg/mL and 13.48 μg/mL, respectively, surpassing both the WT and Δ*vepN* significantly. Additionally, the RT-qPCR results indicate a downregulation of genes involved in aflatoxin biosynthesis in Δ*vepN* at 48 h and 72 h, with *aflR*, *aflC*, *aflO*, and *aflM* being consistently downregulated (Figure 6C–E). Notably, *aflR*, an essential transcriptional regulator, saw a significant decrease at 12 h, profoundly impacting aflatoxin production (Figure 6C).

### 2.8. vepN Is Necessary for A. flavus Peanut Infections

The infection capacity of the Δ*vepN* mutant was markedly diminished, specifically in terms of conidia count, infection area, and AFB content. After 7 days of cultivation, the infection area of Δ*vepN* was notably smaller compared with the WT, with a fifty percent reduction in the conidia count (Figure 7A). The TLC analysis revealed the absence of AFB in peanuts infected by Δ*vepN* (Figure 7C). Over a 14-day culture period, severe damage to the peanut tissue was evident (Figure 7B), with a substantial increase in conidia production in the WT, reaching 1.46 × 10^9^ spores/mL, while Δ*vepN* exhibited slower growth, reaching 4.12 × 10^8^ spores/mL (Figure 7E), and a minimal amount of AFB_1_ synthesis was detected (Figure 7D).

### 2.9. Responses to Different Stressors

The growth of both the wild type (WT) and Δ*vepN* strains was impeded when they were exposed to various stressors in GMM and YES media, as depicted in Figure 8. Under hypertonic stress, Δ*vepN* exhibited heightened sensitivity compared with WT when subjected to 1 M NaCl. In contrast, the inhibitory effect on growth in YES medium with 1.2 M sorbitol was less pronounced, despite the enhancement effect observed on GMM medium. Notably, Δ*vepN* faced significant growth inhibition under oxidative stress conditions, whether in GMM or YES medium treated with 5 mM H_2_O_2_, and was completely inhibited in GMM medium. At 42 °C, Δ*vepN* showed a more pronounced growth inhibition rate compared with WT.

### 2.10. Effects of vepN on the Cell Wall

In the WT strain, the cell wall exhibited uniformity and was encased by a complete fibril layer with intact septa observed (refer to Figure 9A–C). Conversely, in the Δ*vepN* strain, varying levels of thickening were observed in the cell walls (Figure 9D), where the fibril layer appeared somewhat disrupted. Furthermore, the septum exhibited incomplete formation and partial blockage (Figure 9E,F), without complete segregation, potentially impacting intercellular communication.

## 3. Discussion

The gene *vepN* contains a highly conserved GTP-binding domain [22]. The Walker A motif sequence is GAPGTGK[T/S], forming a p-loop that can directly interact with nucleotides [23,24]. The G3 and G4 motifs are responsible for Mg^2+^ binding and imparting specific binding affinity for GTP over other nucleotides. It acts as a molecular switch, regulating various biological processes. This study revealed that *vepN* has a favorable impact on reproduction in *A. flavus*. A primary reproductive mode in this species is asexual reproduction, which is primarily carried out through asexual spores known as conidia [25], which are under the regulation of numerous genes. The transcription factor BrlA, containing a C_2_H_2_ zinc finger DNA-binding domain, plays a crucial role in initiating conidiophore formation and functions predominantly in the prophase. Upon analyzing Δ*vepN*, significant downregulation of *brlA* expression was observed, influencing the activation of the downstream genes *abaA* and *wetA* (refer to Figure 3G). The *wetA* gene is essential for conidia maturation and contributes to the synthesis of conidial wall components such as the inner C4 layer [26,27,28]. The genetic cascade centered around *brlA*→*abaA*→*wetA* and the activation of complementary genes are disrupted [29], resulting in a drastic decrease in conidia count and morphological abnormalities and impacting the infection capability of *A. flavus* (Figure 3B,D–F). Another mode of reproduction in *A. flavus* involves sexual reproduction, where the sclerotial structure is formed by mycelia branching into a dense network, leading to the development of white immature sclerotia that mature with pigment accumulation. The matured sclerotia serve as dormant bodies capable of withstanding adverse environmental conditions until favorable conditions for resuscitation emerge, thereby increasing conidia production [30,31]. NsdC and NsdD encode GATA-type transcription factors, and their mutations result in the absence of sclerotia formation, while the gene *sclR* promotes hyphal fusion [32]. In Δ*vepN*, the relative expression levels of *nsdC*, *nsdD*, and *sclR*, were notably downregulated. Thus, *vepN* positively influences the formation of conidia and sclerotia, facilitating resistance to adverse environments and the occurrence of beneficial mutations (refer to Figure 8). Given the genetic correlation between morphogenesis and secondary metabolism, it is hypothesized that reduced conidia and sclerotia production may also impact aflatoxin production [33].

AflR harbors a GAL4-type binuclear zinc finger cluster domain and serves as a crucial transcriptional regulator within the aflatoxin biosynthetic cluster [34]. Prior investigations have elucidated the binding motif of AflR as 5′-TCG(N5)CGA-3′ and have evidenced its broad functionalities in *A. flavus* growth and development [35]. In our study, within the Δ*vepN* strain, a notable decrease in the relative expression of *aflR* at 12 h was observed, leading to the inference that *vepN* impacts other structural genes by downregulating *aflR*. This downregulation further led to the inhibition of *aflD*, *aflO*, and *aflQ* expression, consequently impeding the formation of various metabolites, such as stericillin and hydroxymethyl stericillin. Within the promoter region of the gene polyketide synthase (*pksA*), two binding sites for AflR were identified, crucial for *pksA* activity [36,37]. Moreover, a significant reduction in the relative expression of the gene *aflC* (*pksA*) was noted. The RNA transcription of the *aflM* gene, critical for the conversion of VERA to demethylated chromatin (DMST), was also found to be repressed [38,39]. Notably, the *aflS*/*aflR* ratio was closely correlated with aflatoxin production, showing a substantial decrease in mutant strains at later stages.

Our findings indicate that *vepN* is associated with stress responses, as its loss reduces the ability to resist against adverse environmental conditions. Additionally, qPCR analysis revealed a decrease in the expression levels of *veA* and *laeA* (another member of the Velvet protein family) (shown in Appendix A), and this study elucidates the regulatory role of *veA* on *vepN*, emphasizing its significance in the growth, development, aflatoxin synthesis, and stress tolerance of *A. flavus*.These novel insights hold essential implications for biological control strategies and genetics.

## 4. Materials and Methods

### 4.1. Strains, Media, and Growth Conditions

The *A. flavus* strains utilized in this study are detailed in Table 1. All strains underwent cultivation at 30 °C in the absence of light. Specifically, the strain TJES19.1 was inoculated in 5/2 agar medium (comprising 5% V-8 juice and 2% agar; pH 5.2) supplemented with 2 mg/mL uracil (Solarbio, Beijing, China) for activation and subsequently incubated at 30 °C for a duration of 5 to 7 days. Conidia were harvested from the 5/2 agar (Solarbio, Beijing, China) by utilizing 0.1% Triton X-100 solution (Sinopharm, Beijing, China) and then introduced into 200 mL of Czapek–Dox broth (Sinopharm, Beijing, China) enriched with 2 mg/mL uracil for transformation experiments. Moreover, in a subsequent phase of the investigation, the cultures were propagated on potato dextrose agar (PDA; Solarbio, Beijing, China), glucose minimal medium (GMM), and yeast extract medium (YES medium).

### 4.2. CHIP-Seq Experiment

The experimental methodology was partly adapted from Kong et al. [40]. In summary, Step 1 involved expressing *A. flavus* VeA (AfVeA; accession number: ABC41691.1; NCBI) and preparing antibodies. The plasmid pET32a (+) harboring *A. flavus veA* was utilized to transform *E. coli* BL21 cells, and its expression was induced with IPTG. AfVeA presence was confirmed in cell lysates through sodium dodecyl sulfate polyacrylamide gel electrophoresis (SDS-PAGE) and Coomassie blue staining, followed by purification using a His-tag. Monoclonal antibodies against AfVeA were generated in mice by Abcam (Shanghai, China), with Western blot hybridization being utilized for specificity verification. In Step 2, chromatin immunoprecipitation (ChIP) was conducted, where 1 × 10^6^ spores/mL were cultivated in potato dextrose broth (PDB; Solarbio, Beijing, China) at 28 °C for 24 h, followed by cross-linking, DNA sonication, and chromatin immunoprecipitation of the resultant cultures. The sequencing of immunoprecipitated DNA was then performed by utilizing the Millipore Chromatin Immunoprecipitation Assay Kit (17-295; EMD Millipore Corporation, Temecula, CA, USA) for ChIP experiments. Step 3 involved the analysis of binding motifs. The generation of ChIP-seq libraries, ChIP-sequencing, and identification of peaks were executed by Berry Genomics (Beijing, China), with the results demonstrating combined reads from the three replicate samples. This research project was reviewed by the Experimental Animal Welfare and Ethics Committee of Hangzhou Yuhang Keli Rabbit Industry Professional Cooperative and complies with the principles of experimental animal welfare ethics. The ethical approval number is KLTYLAC-20161001-01.

### 4.3. Phylogenetic Analysis

*vepN* orthologs were obtained from sequences sourced from the National Center for Biotechnology Information (NCBI) website (https://www.ncbi.nlm.nih.gov/, accessed on 20 January 2023). The sequences were subjected to BLAST analysis to identify the microfungal species with the highest percentage similarity in the database. Alignment of the DNA sequences was carried out by using MEGA 11 (version 11, Mega Limited, Auckland, New Zealand) for subsequent phylogenetic analysis of the isolates via the maximum likelihood method with bootstrap testing consisting of 1000 replications. The phylogenetic tree depicting the relationships among the 13 VepN orthologs was constructed using the MEGA 11 algorithm.

### 4.4. Construction of Gene Knockout and Overexpression Strains

The gene knockout strategy by double-crossover recombination has been described in detail elsewhere [40]. A fusion PCR-based method was used to construct the Δ*vepN* strain, as depicted in Figure 3A. The two flanking fragments 5′UTR and 3′UTR of the gene *vepN* and the screening marker gene *pyrG* were amplified, respectively, using Phanta Max Super-Fidelity DNA polymerase (Vazyme Biotech Co., Ltd., Nanjing, China). TJES19.1 was used as the host for transformation, and the obtained transformation mixture was cultured in CZ regeneration medium supplemented with 0.6 M KCl for 7 to 10 days. The obtained transformants were validated by PCR and sequencing. 5′UTR, *SH ble* (phleomycin resistance gene) [41], *gpdA*, *vepN*, 3′UTR, and the vector PUC19 were amplified by using specific overlapping primers. A 1:1 molar ratio between the vector and each fragment was used and assembled by using the Gibson Assembly Cloning Kit (New England Biolabs, Ipswich, MA, USA). Δ*vepN* was used as the transformed host, and 30 μg/mL phleomycin (Maokang Biotechnology Co., Ltd., Shanghai, China) was added to the regeneration medium for primary screening, while a 100 μg/mL concentration was used for secondary screening. Positive transformants were verified by PCR and sequencing (sequencing was entrusted to Sangon Biotech, Shanghai, China). Primers used instrategy and confirmation of the mutant strain and complemented strain are shown in Appendix A and Appendix A respectively.

### 4.5. Observation of Conidia Morphology and Colonies

Approximately 3 μL aliquots of a 10^8^ spores/mL suspension of the WT, Δ*vepN*, and *vepN*-com strains were point-inoculated into GMM, PDA, and YES media and incubated at 30 °C in the dark. Conidia morphology was assessed with a microscope (Olympus BX41; Tokyo, Japan) after 48 h for cultures on GMM medium, while colony morphology was evaluated after 5 days for cultures on all media. (Three replicates were obtained for each strain.)

### 4.6. Conidial Formation

Aliquots of 3 μL of a 10^8^ spores/mL suspension of the WT, Δ*vepN*, and *vepN*-com strains were point-inoculated at the center of glucose minimal medium (GMM), potato dextrose agar (PDA), and yeast extract sucrose (YES) media. The plates were then incubated at 30 °C for 3, 5, and 7 days. After the respective incubation periods, three 1.5 cm diameter agar plugs were cut from each plate, transferred to a 10 mL centrifuge tube (Sangon Biotech, Shanghai, China), and treated with 5 mL of a 0.1% Tween 80 solution (Sinopharm, Beijing, China) before being vortexed for approximately 10 min. Subsequently, the agar plugs were removed, and the conidia were quantified by using a hemocytometer (Solarbio, Beijing, China) with a microscope (Olympus BX41; Tokyo, Japan). This procedure was performed in triplicate for each strain.

### 4.7. Sclerotial Production

Aliquots of 3 μL of a 10^8^ spores/mL suspension of the WT, Δ*vepN*, and *vepN*-com strains were point-inoculated onto GMM agar supplemented with 2% sorbitol (Sinopharm, Beijing, China) and incubated at 30 °C in the dark for 7 and 14 days [33]. The sclerotia were treated with 75% ethanol, observed, and counted in various areas. Each experiment was replicated three times per strain.

### 4.8. Determination of Aflatoxin

(i) Thin-layer chromatography (TLC) was employed for aflatoxin analysis. Aliquots of 3 μL of a 10^8^ spores/mL suspension of the WT, Δ*vepN*, and *vepN*-com strains were point-inoculated on GMM, PDA, and YES media and incubated for 5 days, followed by testing for aflatoxin B_1_ and aflatoxin B_2_ (AFB) contents [31]. Aflatoxin analysis was conducted as follows: Three 1.5 cm diameter agar plugs per plate were transferred to 50 mL centrifuge tubes (Sangon Bitech, Shanghai, China), treated with 5 mL of methanol (Sinopharm, Beijing, China), vortexed for 20 min, and shaken at 200 rpm for 1.5 h. The resulting supernatant was relocated to new 10 mL centrifuge tubes post-centrifugation by a high-speed freezing centrifuge (Shanghai Anting Science, Shanghai, China), evaporated by a drying oven (shanghai longyu biotechnology co. ltd, Shanghai, China), and redissolved in 100 μL of CHCl_3_ (Sinopharm, Shanghai, China). Subsequently, the extract was vortexed, and 10 μL was applied to a TLC plate (Merck KGaA, Darmstadt, Germany) (each aliquot being 1 cm apart and 2 cm away from the edges), running in a solvent mixture of toluene–ethyl acetate–acetic acid (65:35:10 [*v*/*v*/*v*]). AFB was visualized under 365 nm UV light. Each strain was tested in triplicate.

(ii) High-performance liquid chromatography (HPLC) was utilized for further aflatoxin quantification. Cultures on YES medium were incubated at 30 °C for 5 days, and aflatoxin content was extracted by using methanol and purified through an immunoaffinity column. The eluted extract underwent filtration with a 0.22 µm filter (Millipore, Darmstadt, Germany) following elution from the immunoaffinity column with 1 mL of HPLC-grade methanol (Merck KGaA, Darmstadt, Germany). An AFB standard from Sigma, USA, dissolved in HPLC-grade methanol at a concentration of 1 mg/mL was kept at 4 °C in the dark until analysis. Separation was carried out using an Agilent 20RBAX Eclipse XDB-C18 column at 30 °C, with a mobile phase comprising ultrapure water, acetonitrile, and methanol (60:20:20 [*v*/*v*/*v*]) at a flow rate of 1 mL/min. Sample injection was conducted accordingly. Each strain was analyzed in triplicate.

### 4.9. Seed Infection

Mature peanuts underwent surface sterilization with 75% ethanol. A suspension of 10^5^ spores/mL was inoculated into a 250 mL Erlenmeyer flask (Sangon Bitech, Shanghai, China) containing 20 peanut cotyledons and incubated at 30 °C with 90% humidity for 7 and 14 days. Subsequently, the infected peanuts were transferred to a 50 mL centrifuge tube with the addition of 5 mL of methanol, followed by vortexing for 20 min and shaking at 200 rpm for 1.5 h. The resultant mixture was utilized for conidia examination by using a hemocytometer, and the supernatant was employed for AFB analysis via TLC. Each experimental condition was replicated three times.

### 4.10. Stress Experiment

Aliquots of 3 μL of 10^8^ spores/mL of both the WT and Δ*vepN* strains were inoculated into GMM and YES media under distinct stress conditions at 30 °C for 4 days. Post-incubation, colony growth was assessed. Cell wall stress involved exposure to 100 μg/mL sodium dodecyl sulfate (SDS) and 300 μg/mL Congo red (CR), hypertonic stress comprised 1 M NaCl and 1.2 M sorbitol, and oxidative stress entailed 5 mM H_2_O_2_. High-temperature stress was induced at 42 °C. (Each strain underwent three replicate experiments.)

### 4.11. Transmission Electron Microscope (TEM)

A suitable fresh conidial solution was introduced into 20 mL of GMM broth and incubated at 30 °C for 48 h. Following the growth period, the mycelial pellets underwent three washes in distilled water and were then fixed in 2.5% glutaraldehyde for the preparation of ultra-thin sections. Subsequent imaging was conducted by using a JEM-1200EX electron microscope (Japan Electronics Corporation, Tokyo, Japan), and the images were scanned for in-depth analysis of the cell walls of both the WT and Δ*vepN*. (Experiments for each strain were replicated thrice.)

### 4.12. RT-qPCR Analysis

First, we performed total RNA extraction. Strains were grown on PDA at 30 °C for 7 days and washed with 0.01% Tween 80 solution; then, they were transferred to PDB and shaken at 180 rpm for 12 h. The mycelia were harvested by using sterilized Miracloth (Millipore, Darmstadt, Germany), washed with sterile water, dried, and laid flat on a PDA agar plate at 30 °C. Samples were collected at various time points (0 h, 6 h, 12 h, 24 h, 48 h, and 72 h) and stored at −80 °C. Secondly, total RNA was extracted by employing the Fungal RNA Kit (Omega Biotek, Inc., Darmstadt, Germany), followed by reverse transcription into cDNA by using All-In-One 5X RT Master Mix (Applied Biological Materials Inc., Vancouver, Canada). Lastly, RT-qPCR was conducted in a 20 μL reaction volume with Hieff^®^ qPCR SYBR Green Master Mix (Low Rox Plus; Yeasen Biotechnology Co., Ltd., Shanghai, China). Gene expression levels were compared relative to a level of 1 set for the WT control, and all gene expression values were normalized to *A. flavus* 18S rRNA. The relative gene expression was determined by using the 2^−ΔΔCt^ method. Primers used in RT-qPCR and sequencing are shown in Appendix A. (Each strain was analyzed in triplicate.)

### 4.13. Statistical Analysis

Quantitative data were analyzed by using analysis of variance (ANOVA) in combination with Tukey’s post hoc test with SPSS. Statistical significance was determined at *p* < 0.05, indicating significant differences among the average comparisons of *Aspergillus flavus* strains.

## Figures and Tables

**Figure 1 toxins-16-00174-f001:**
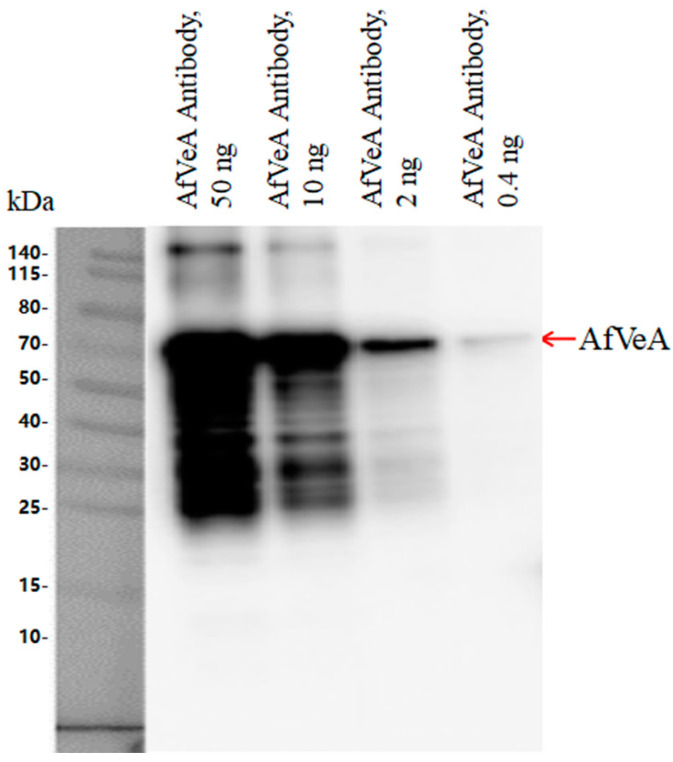
Western blot analysis demonstrating specificity of AfAflR antibody.

**Figure 2 toxins-16-00174-f002:**
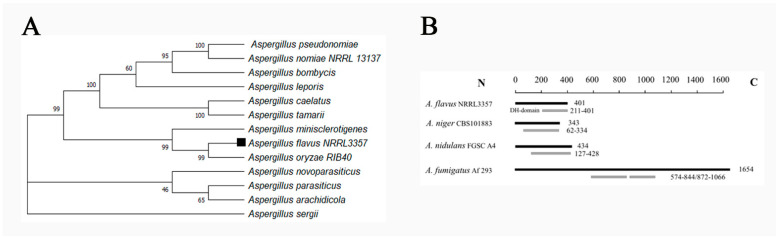
Characterization of *vepN* in *A. flavus*. (**A**) Phylogenetic relationship of VepN homologs from different species analyzed with MEGAX 11. (**B**) Domain architecture of P-loop NTPase. Numbers indicate positions of amino acid residues in protein domains.

**Figure 3 toxins-16-00174-f003:**
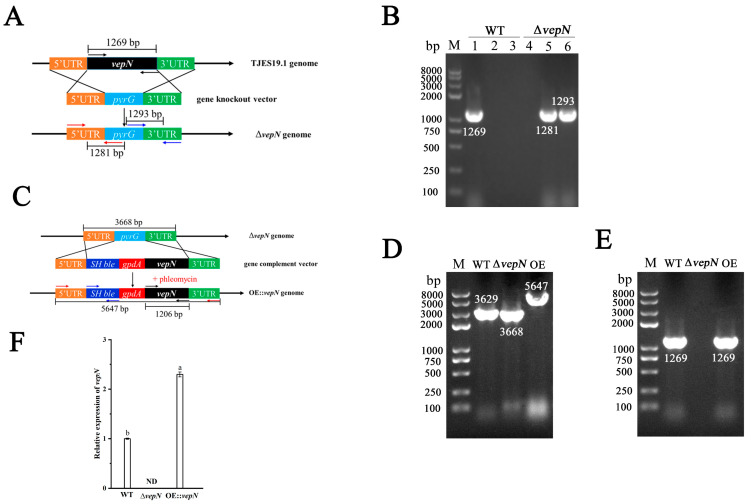
The strategies and confirmation of the mutant strains. (**A**) The scheme of the *vepN* deletion strategy. Black arrow: target gene verification; Red arrow: upstream verification; Blue arrow: downstream verification. (**B**) Gene knockout was verified by PCR analysis. M: maker; Lanes 1 and 4: gene *vepN*; Lanes 2 and 5: upstream verification; Lanes 3 and 6: downstream verification. (**C**) The scheme of the *vepN* overexpression strategy. Black arrow: target gene verification; Red arrow: OE vector verification; Blue arrow: *SH ble* gene. (**D**) The OE strain was verified by PCR analysis of the upstream and downstream regions. (**E**) The OE strain was verified by the PCR analysis of the gene *vepN*. (**F**) The expression analysis of *vepN* by qRT-PCR indicating greater accumulation of *vepN* transcripts in the *vepN* overexpression strain compared with the wild type. The strains were cultivated on PDA at 30 °C for 7 days and then transferred to PDB and cultured for 12 h. The relative expression was calculated using the method of 2^−ΔΔCt^. The expression of 18S rRNA was used as an internal reference. The values were normalized to the expression levels in the wild type, considered to be 1. The error bars represent the standard errors, and different letters above the bars represent significantly different values (*p* < 0.05).

**Figure 4 toxins-16-00174-f004:**
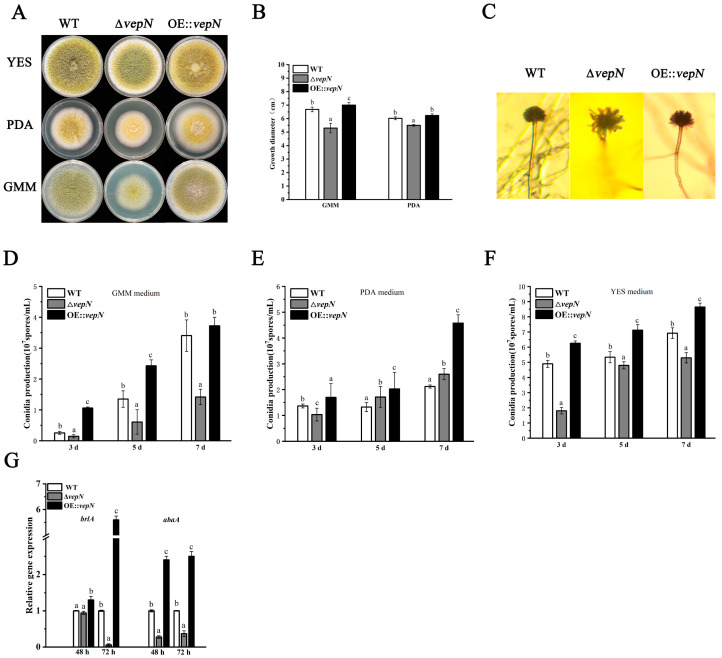
The roles of *vepN* in conidiation in *A. flavus*. (**A**) Colony morphology on different media (GMM, PDA, and YES) cultured for 5 days. (**B**) Microscopic observation of conidiophores on GMM medium at 48 h. (**C**) Growth diameter of strains when cultured on GMM and PDA media for 5 days. Conidial production of WT, Δ*vepN*, and OE::*vepN* strains on different media: GMM medium (**D**), PDA medium (**E**), and YES medium (**F**). (**G**) The q-PCR analysis of the genes *brlA* and *abaA* for conidiation. The error bars represent the standard errors, and different letters above the bars represent significantly different values (*p* < 0.05).

**Figure 5 toxins-16-00174-f005:**
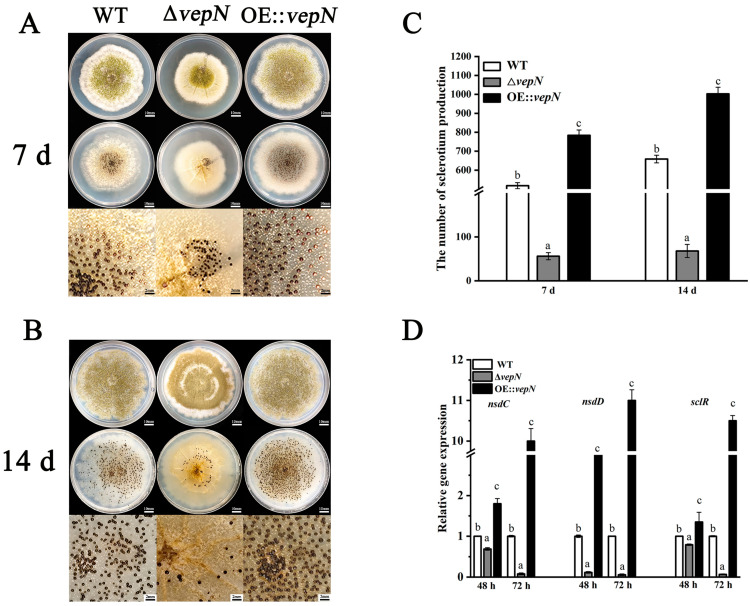
The roles of *vepN* in sclerotia formation in *A. flavus*. Sclerotial development at (**A**) 7 and (**B**) 14 days. Upper panels: colonial morphology. Middle panels: exposed sclerotia after the mycelium and conidia were washed off. Bottom panels: closeups of the central area that show the sclerotia produced. For sclerotial development and production on GMM medium with 2% sorbitol, cultures were grown in the dark at 30 °C for 7 and 14 days; then, the plates were sprayed with 75% ethanol to make the sclerotia visible, and these were counted in various areas. (**C**) The number of sclerotia counted at 7 days and 14 days. (**D**) The q-PCR analysis of *nsdC*, *nsdD*, and *sclR*, which are the genes of the positive regulators of sclerotia formation. The error bars represent the standard errors, and different letters above the bars represent significantly different values (*p* < 0.05).

**Figure 6 toxins-16-00174-f006:**
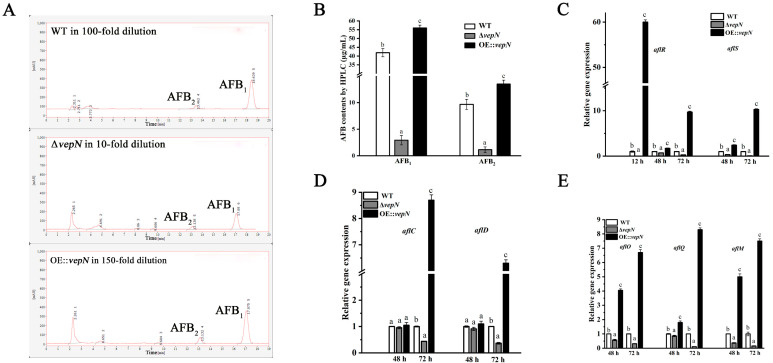
The roles of *vepN* in aflatoxin biosynthesis in *A. flavus*. (**A**,**B**) Liquid chromatography (HPLC) was used to determine the AFB profiles of the WT, Δ*vepN*, and OE::*vepN* grown on YES medium at 30 °C for 5 days. (The liquid chromatography profile shows that the three strains were diluted by different multiples.) (**C**–**E**) The q-PCR analysis of aflatoxin biosynthesis-related genes *aflR*, *aflS*, *aflC*, *aflD*, *aflO*, *aflQ*, and *aflM*. The error bars represent the standard errors, and different letters above the bars represent significantly different values (*p* < 0.05). (Three replicates were obtained for each strain.)

**Figure 7 toxins-16-00174-f007:**
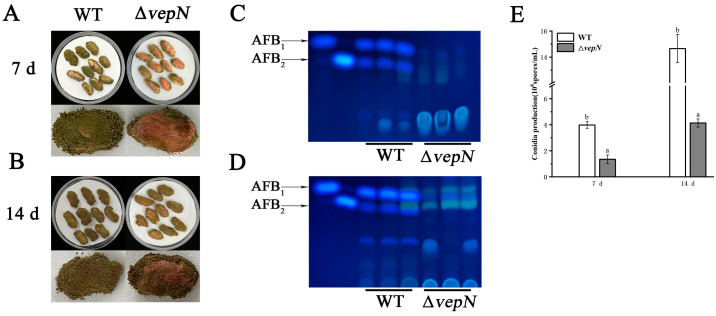
The role of *vepN* in the *A. favus* pathogenicity against peanut seeds. The infection of peanuts by the WT and Δ*vepN* at 7 (**A**) and 14 days (**B**). The production of AFB from peanuts infected by the WT and Δ*vepN* was detected by TLC at 7 (**C**) and 14 days (**D**). (**E**) Conidia production in infected peanuts. The error bars represent the standard errors, and different letters above the bars represent significantly different values (*p* < 0.05). (Three replicates were obtained for each strain.)

**Figure 8 toxins-16-00174-f008:**
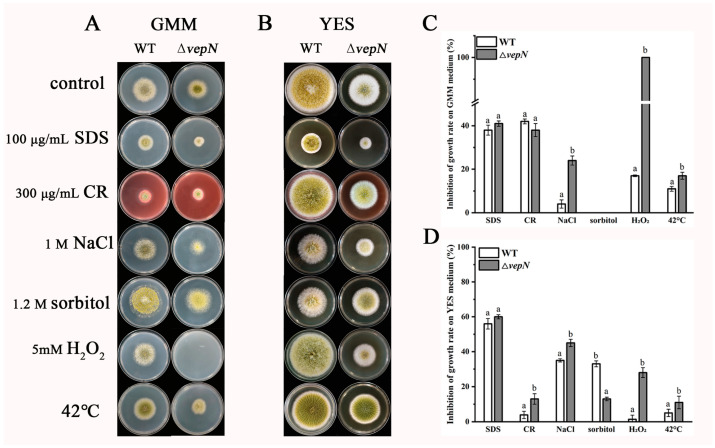
The roles of *vepN* in response to external stress in *A. flavus*. Colony growth morphology on GMM medium (**A**) and YES medium (**B**) under different stress conditions at 30 °C for 4 days. (**C**) The inhibition of the growth rate on GMM medium. (**D**) The inhibition of the growth rate on YES medium. The growth inhibition rate was calculated by measuring the colony diameter. The error bars represent the standard errors, and different letters above the bars represent significantly different values (*p* < 0.05). (Three replicates were obtained for each strain.)

**Figure 9 toxins-16-00174-f009:**
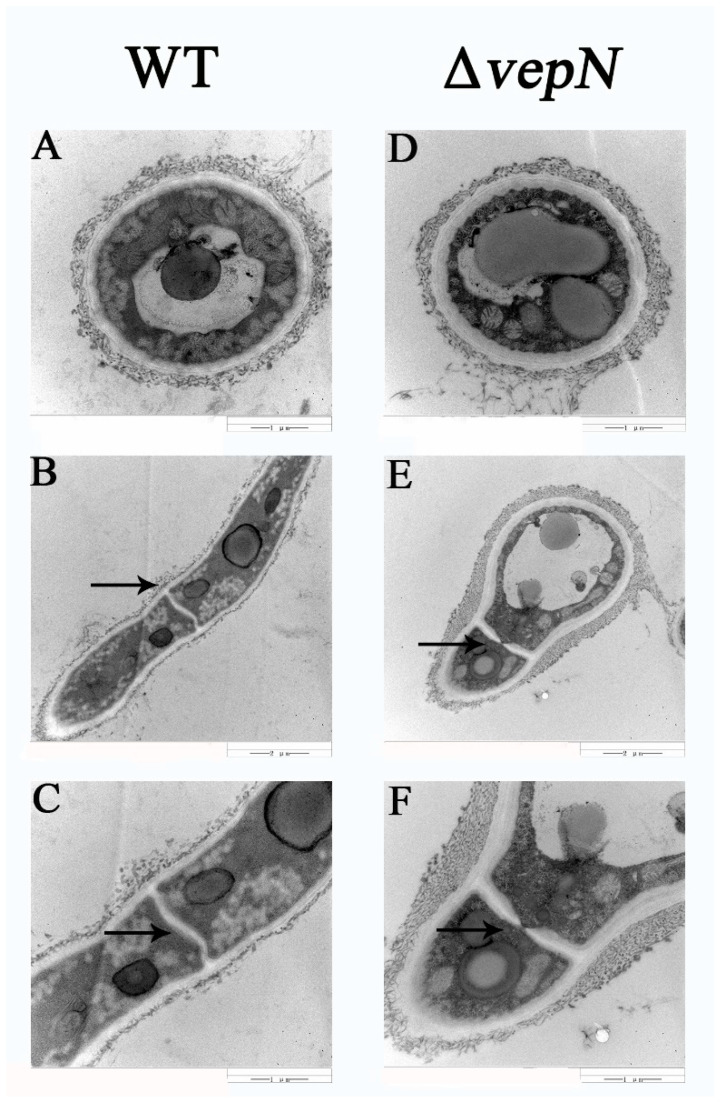
The effect of *vepN* on the cell wall in *A. flavus*. The cross section (**A**,**D**). The arrowheads indicate the regular septa (**B**,**C**) and blocked septa (**E**,**F**) observed in the vertical section.

**Table 1 toxins-16-00174-t001:** *A. flavus* strains utilized in this study.

Strains	Description
WT	*A. flavus* NRRL3357 wild type
TJES19.1	Δ*Ku70* and Δ*pyrG*
Δ*vepN*	Δ*Ku70*, Δ*vepN*, and *pyrG*^+^
OE::*vepN*	Δ*Ku70*, Δ*pyrG*, *vepN*^+^, *gpdA^+^*, and *SH ble^+^*

## Data Availability

Dataset available on request from the authors.

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
