# Peer review of "The Gene vepN Regulated by Global Regulatory Factor veA That Affects Aflatoxin Production, Morphological Development and Pathogenicity in Aspergillus flavus"

_toxins, 2024, doi:10.3390/toxins16040174_

Round 1

Reviewer 1 Report

Comments and Suggestions for Authors

The authors present the manuscript with the title „The Gene vepN Regulated by Global Regulatory Factor veA that 2 Affects Aflatoxin Production, Morphological Development and 3 Pathogenicity in Aspergillus flavus”.

The manuscript is very well presented, and I have only 4 minor aspects that I recommend to be revised:

Line 87. Replace , with .

Lines 263-265. The results of the present study allow the authors to make this statement, is it just a supposition/speculation or is there a link between the statement and other previously published data (according to the cited bibliography). Please clarify this aspect.

Line 293. I recommend introducing a new section (4.1) with the title Chemicals, reagents and supplies. In this section information about the manufacturer and distributor for all reagents (eg agar, uracil, ethanol, tween 80, methanol, etc.) and other materials (such as centrifuge tubes, syringe filters, immunoaffinity column, etc.) should be introduced.

Line 360. Mention the equipment for centrifugation and evaporation.

Reviewer 2 Report

Comments and Suggestions for Authors

General:

Probably all the figures need to be larger.  Possibly much larger, or if they must be this small, at least high enough quality so that detail can be seen when the image is enlarged.  Figure 5B is especially bad.  

It is very frustrating to see aflatoxin research being done today that relies on YES or other synthetic liquid media.  Quoting from Probst and Cotty (2012) who explicitly tested YES media: “There was no correlation between aflatoxin production in viable maize and production in any of the tested liquid fermentation media using any of the fermentation techniques. ”  I am inclined to reject manuscripts that continue to disregard this finding, but I’ll make an allowance here because the conclusions are not exclusively predicated on results with YES.  

Specific points:

Abstract “weird” conidial heads is too casual / imprecise.  Just state what the difference is

Section 2.5 / Line 142-151:  This is quantitative data.  I’d encourage you to be specific in your discussion here.  For example say “an eight-fold increase” or “over 30% more”.  You frequently state that something is “significant” but it is not clear that this was actually examined statistically. 

Figure 4:  move the placement so the caption does not extend onto the next page.

Figure 4:  If possible, please include a scale bar for the bottom panel on 4A and 4B.  

Figure 4:  I don’t see the value in both TLC and HPLC results.  I’d suggest presenting just one or the other.  

Figure 4:  Is the green spot just below the B2 spot sterigmatocystin?

Figure 7:  Capitalize the C in NaCl

Line 337:  Suggest wording “…were point inoculated on GMM…” 

Line 352:  Not clear how the sclerotia were counted.

Reviewer 3 Report

Comments and Suggestions for Authors

Authors have presented a gene study on the gene vepN in Aspergillus flavus. The study has demonstrated that this gene has a significant impact on regulating morphological development, secondary metabolism, and virulence of A. flavus. While initially identified by association with VeA, no further mention of this connection is discussed despite the role VeA plays as a global regulator of many of the facets the author investigates. This lack of discussion begs the question why did you not look at this? At the very least expression of veA and other velvet members could be assessed by the cDNA you already created.

The data presented by the authors has merit but is underwhelming in presentation quality as some information critical to understanding of the figures is missing or the images themselves are of low quality and thus hard to even read. For example, I am unable to read any of labels on many of the graphs to fully understand what is being presented such as with Figure S1.

In the results section detailing the GO analysis more information needs to be provided about the CHIPseq experiment so that proper context can be given to the data. The information about the conditions of the experiment is provided in the materials and methods but should be summarized in the results section.

An error that occurs first in figure 3 but occurs in multiple other figures is that no explanation of what media condition used for gene expression is given. The methods indicate that this was done on PDA but also mention other timepoints and conditions that were assessed not shown. Based on what is written in the methods and shown in the figures, the gene expression data does not correspond with the morphological data presented alongside it. This information should be clearly stated so that readers do not imply the wrong conclusions.

In Figure 2 labels or descriptions defining what type of PCR reaction run in each well is absent. While I was able to figure out what was tested by looking back at the main text, all relevant information should be included in the figure legends. Also, I suggest changing the word line to lane for clarity. Figure 2D needs negative controls included to show how “line 1” & “line 2” confirm the complementation is correct.

In your analysis of the asexual development regulators, why did you not examine wetA? You mention this gene in your discussion but don’t examine it with your gene expression.

The HPLC image in Figure 5B is not labeled so that we can identify any of the peaks.

With the results of the gene expression on aflatoxin production, at line 193, authors mention aflR being down regulated at 12 h. Figure 7D only shows expression for 48 and 72 h timepoints. Is the previous statement incorrect or is there missing data?

No explanation of what type of statistics ran are explained in the paper. Please add this information at the relevant points or make note of it as a section in the materials and methods.

Text mentions used of MEGA5 vs figure legend stating use of MEGA10 for figure 1. Currently the software is on version 11 so I would like to see the use of a more recent version than version 5. Additionally, no information is provided on what type of analysis was performed in MEGA as well as there is no reference provided for the software.

Based on data from FungiDB and NCBI your genes appear to encode for a transporter of some type. I would double check that your structure analysis is up to date as possible.

I am also very confused on your complementation. What is the purpose of using the constitutive promoter in your complementation? You place the gene back in the native locus so why would you need a new promoter? Additionally, this promoter should cause the gene to be expressed at higher levels than normally. Please demonstrate the expression of the strain used in your study and show the expression of vepN. Since you have not done any gene expression analysis with the complement strain, despite using in most experiments, it leads me to believe that the strain’s expression is not like WT. What reason was this control not included in any of the gene expression analysis? If the data exists, please include it!

Other Comments:

The sentence starting on line 54 seems very out of place and does not lead into the conclusion on the following sentence. I would remove it as the protein being associated VeA is a strong indicator of it having a role in secondary metabolism and aflatoxin production.

For the virulence data, authors mention the kernels being “damaged”. I am not sure this best description of the image and describing the colonies as more fully colonized by the WT vs the mutant would be a more appropriate description. Also why is the complementation not included in this study?

Line 102-103 mis references Figure 2A as Figure 3A

Table 1 should include the phleomycin resistance gene as required.

Comments on the Quality of English Language

The article as it stands is understandable does not have any major issues in language it does contain numerous typos, several occurrences of awkward phrasing and grammatical errors. 

Having an language editor go through the document would enhance the overall quality of the work.

Round 2

Reviewer 3 Report

Comments and Suggestions for Authors

Authors have addressed majority of my concerns and have made great improvements to the paper. I would have preferred the authors test wetA gene expression directly but based on the conclusion the authors have made in the paper actual it is not required.

My only remaining complaint is still with Figure S1. The label are still to hard to read for a publication quality figure. At this point I am not sure if this an issue with the version I am working with that would be corrected in the published document but I ask authors to double check this. 
